# Branched-Chain Amino Acid Supplementation Enhances Substrate Metabolism, Exercise Efficiency and Reduces Post-Exercise Fatigue in Active Young Males

**DOI:** 10.3390/nu17071290

**Published:** 2025-04-07

**Authors:** Chenglin Luan, Yizhang Wang, Junxi Li, Nihong Zhou, Guilin Song, Zhen Ni, Chunyan Xu, Chunxue Tang, Pengyu Fu, Xintang Wang, Lijing Gong, Enming Zhang

**Affiliations:** 1Key Laboratory of Exercise and Physical Fitness of Ministry of Education, Beijing Sport University, Beijing 100084, China; luanchenglin59@gmail.com; 2China Institute of Sport and Health Science, Beijing Sport University, Beijing 100084, China; wangyizhang250@gmail.com (Y.W.); lijunxi@bsu.edu.cn (J.L.); nizhen888@bsu.edu.cn (Z.N.); 3School of Sport Science, Beijing Sport University, Beijing 100084, China; zhounihong@bsu.edu.cn (N.Z.); xucy@bsu.edu.cn (C.X.); tcx_19910201@163.com (C.T.); 4Beijing Competitor Sports Science &Tech. Co., Ltd., Beijing 102299, China; songguilin@chinacpt.com; 5Department of Physical Education, Northwestern Polytechnical University, Xi’an 710072, China; fupy@nwpu.edu.cn; 6Key Laboratory for Performance Training & Recovery of General Administration of Sport, Beijing 100084, China; 7Department of Clinical Sciences in Malmö, Lund University Diabetes Centre, Lund University, 20213 Malmö, Sweden; enming.zhang@med.lu.se

**Keywords:** BCAA, substrate metabolism, exercise performance, fat oxidation, CHO oxidation

## Abstract

**Background:** Branched-chain amino acids (BCAAs, isoleucine, leucine, and valine) are commonly applied to promote muscle protein synthesis. However, the effects of BCAAs on exercise-induced substrate metabolism, performance and post-exercise fatigue during endurance exercise remain unclear. **Methods:** In a double-blind cross-over design, eleven active males completed 1 h of constant load exercise (CLE) at 60% VO_2_max power followed by a time to exhaustion (TTE) test at 80% VO_2_max power after supplementation with BCAAs or placebo on consecutive three days. During exercise, indirect calorimetry was used to measure the carbohydrate (CHO) and fat oxidation rate, as well as the cycling efficiency. In addition, rating of perceived exertion (RPE) and visual analogue scale (VAS) scores were obtained at interval times during the whole period. Fingertips and venous blood (*n* = 8) were collected for the measurement of metabolic responses at different time points during exercise. **Results:** Compared to the placebo group, the fat oxidation rate was significantly higher after 20 and 30 min of CLE (*p* < 0.05). The CHO oxidation rates showed a significant increase in the BCAA group during TTE (*p* < 0.05). Meanwhile, the cycling efficiency during TTE was significantly improved (*p* < 0.05). Interestingly, VAS significantly decreased post-exercise in the BCAA group (*p* < 0.05). Additionally, the levels of blood insulin between the two groups were significantly higher in the post-exercise period compared to the pre-exercise periods (*p* < 0.001), while insulin levels were significantly lower in the post-exercise period with supplemental BCAAs compared to the placebo (*p* < 0.001). BCAAs also enhanced the levels of blood ammonia in the post-exercise period compared to the fasting and pre-exercise periods (BCAA: *p* < 0.01; Placebo: *p* < 0.001). However, in the post-exercise period, blood ammonia levels were significantly lower in the BCAA group than in the placebo group (*p* < 0.05). **Conclusions:** This study shows the critical role of BCAAs during exercise in active males and finds that BCAA supplementation enhanced fat oxidation during the CLE, increased carbohydrate oxidation and exercise efficiency during the TTE, and reduced immediate post-exercise fatigue.

## 1. Introduction

Substrate metabolism plays a key role in endurance exercise performance, and there is a delicate interplay interaction between different substrates during moderate to high-intensity exercise, which in turn affects exercise performance [1]. The main metabolic substrates are carbohydrates (CHO) and fats [2]. Enhanced fat oxidation during prolonged endurance can reduce glycogen depletion [3], which is a key factor in delaying fatigue [4]. Therefore, enhancing fat oxidation has the potential to prolong the duration of exercise by delaying glycogen depletion and the onset of fatigue. Several methods are able to identify the enhancement of fat oxidation during exercise [3]; for example, recent studies show that a low-carbohydrate high-fat diet (LCHF) increases fat oxidation and reduces CHO utilization post-exercise. However, LCHF impairs exercise metabolism and performance [5]. Thus, strategies to enhance fat oxidation during prolonged exercise without compromising metabolism and performance remain a critical challenge.

Branched-chain amino acids (BCAAs) are essential nutrients obtained through dietary intake [6], playing a critical role in energy supply during rest and exercise [7]. After intake, BCAAs bypass hepatic catabolism because of the low activity of the BCAA aminotransferase and directly flow in the blood for use by skeletal muscle or other tissues [8]. In these tissues, a series of enzymatic reactions convert BCAAs into acetyl-coenzyme A (CoA) and succinyl-CoA, which then enter the tricarboxylic acid (TCA) cycle, followed by a series of reactions that produce large amounts of adenosine triphosphate (ATP), providing energy for the needs of the body [9].

Besides energy supply, BCAAs are known to enhance substrate metabolism within the body. BCAA levels in the body are closely related to glucose metabolism [10]. For example, in rats, four weeks of a BCAA diet increased plasma BCAA levels, reduced the activity of the pyruvate dehydrogenase (PDH) complex in the liver, and sustained higher blood glucose levels after exercise [11]. Furthermore, as mentioned in the review, the catabolism of BCAA in the body also affects fat metabolism [12]. For instance, in mice, BCAA combined with high-fat feeding reduces body weight, decreases adipose tissue by inhibiting hepatic lipogenic gene expression, and reduces hepatic lipogenesis and autophagy [13]. In humans, participants exercised to exhaustion at 70% VO_2_max power after ingesting supplemental BCAA (80 mg/kg body weight), which enhances plasma non-esterified fatty acid (NEFA) levels immediately and 30 min post-exercise [14].

Although CHO and fats are the primary energy sources during exercise, amino acid metabolism, particularly the catabolism of BCAA, also plays a crucial role in skeletal muscle during prolonged exercise [15]. BCAA intake during pre-exercise enhances the uptake of BCAA in skeletal muscle during exercise, suggesting that external supplementation exerts direct effects on exercise [16], such as enhancing muscle strength [17], reducing post-exercise immunosuppression [18], alleviating exercise-induced fatigue, and promoting exercise recovery [19]. However, the impact of BCAA supplementation on the endurance exercise performance of humans requires further research.

Central fatigue is a complex phenomenon that occurs due to ammonia accumulation in the blood and alterations in neurotransmitter synthesis [20], particularly an increase in serotonin, which induces a feeling of tiredness during exhaustive exercise [21]. The uptake of the serotonin precursor Tryptophan (Trp) in the brain increases significantly during prolonged sustained exercise [22]. BCAAs can compete with Trp to cross the blood–brain barrier and are potent in mitigating central fatigue during high-intensity exercise. Furthermore, supplementation with BCAAs may also reduce cerebral serotonin synthesis, thereby preventing the onset of central fatigue during prolonged high-intensity exercise [23]. BCAA digestion enhances time trial (TT) performance or extends time to exhaustion (TTE) during prolonged exercise [24,25], also reducing ratings of perceived exertion (RPE) [24] and blood lactate levels [26] during constant load exercise (CLE). However, some studies have indicated that BCAA intake does not improve endurance performance (50%VO_2_max load cycling) [27,28] when attempting to explore the comprehensive role of BCAAs during endurance exercise.

In this study, eleven active male college students were recruited based on strict inclusion criteria. Participants underwent a randomized, placebo-controlled, crossover trial to investigate the effects of BCAA supplementation on substrate metabolism and exercise performance following endurance exercise. The findings show that pre-exercise BCAA supplementation enhances fat oxidation during CLE, improves CHO oxidation and boosts the efficiency of endurance exercise during TTE. Notably, BCAA supplementation significantly alleviates exercise-induced fatigue, highlighting the critical role of BCAA in regulating energy homeostasis and modulating central nervous system function. Therefore, this study provides evidence to support more diverse nutritional strategies during exercise.

## 2. Materials and Methods

### 2.1. Participants

Participants were recruited from healthy, active male college students. Participants were asked if they met the current ACSM activity guidelines of >150 min of vigorous exercise per week. Each participant completed both a physical activity readiness questionnaire (PAR-Q+) and a health information form to minimize any potential contraindications to exercise [29]. Participants were excluded if they had been taking protein or BCAA supplements for the last six months or were suffering from or had a previous history of renal, gastrointestinal, musculoskeletal injury, or cardiovascular complications. Prior to giving informed consent, participants received an information sheet and a verbal outline of the risks and benefits of taking part in the study. Fifteen young male subjects were assessed for eligibility, and 3 subjects were excluded because they did not meet the inclusion criteria. Among the 12 subjects, one subject had their tests interrupted due to injury after randomization. The final sample comprised eleven young males (Table 1). The study was approved by the Beijing Sport University Ethical Committee (No. 2023233H).

### 2.2. Study Design

The study was a double-blind, placebo-controlled within-subject design. Following confirmation of eligibility to partake in the study and the signing of informed consent, the participants underwent body composition testing and a maximal exercise test to exhaustion during power cycling to establish the maximal oxygen consumption and peak power output that was used to calculate the power at 60% and 80% of VO_2_max. Next, randomization was performed (www.randomizer.org, accessed on 1 October 2023) to establish the order of treatments over the experimental visits. Randomization and treatment preparation were conducted by a member not involved in data collection and analysis. Testing took place between October 2023 and December 2023 in the Beijing Sport University Sports Science Research Centre.

The subjects underwent two experimental tests throughout the experimental phase, with seven washout days for each exercise test (Figure 1). After grouping, the subjects were supplemented with BCAA or a placebo for three consecutive days before the exercise experiment [30]. On the day of the exercise experiment, subjects were required to fast for 8 h before visiting the laboratory for fingertip and venous blood collection. Next, the subjects were asked to consume the supplementations and rest for 30 min. Then, second-time fingertip and venous blood samples were collected for pre-exercise [31]. Exercise experiments were performed at the end of blood collection, with a protocol of 1 h of 60% VO_2_max power constant load exercise (CLE) and 80% VO_2_max power time to exhaustion (TTE). During CLE, the subjects’ gases were collected and their heart rate (HR) was recorded every 10 min. The subjects’ Rating of Perceived Exertion (RPE) and Visual Analogue Scale (VAS) were recorded every 15 min, and fingertip blood was also collected to measure their blood glucose and blood lactate every 15 min. Gases were collected during all the TTE periods. After TTE, the VAS was recorded, venous blood was collected immediately, and fingertip blood was collected on the 1st, 5th, 9th min post-exercise. The washout period was 7 days.

### 2.3. Body Composition and Maximal Aerobic Capacity Assessment

The subjects’ height, weight and body composition were measured using dual-energy X-ray absorptiometry (iDXA, Chicago, IL, USA). Next, an incremental load test to exhaustion on a cycle ergometer (Monark 839E, Vansbro, Sweden) was conducted to measure VO_2_max and the power at VO_2_max. Following a 5 min warm-up at 50 W and rest for 3 min, power output increased by 40 W every 3 min. Power output increased by 20 W every 2 min when the power output reached 200 W. Participants were asked to cycle at a self-selected cadence but to keep this above 60 rpm. The test was terminated upon volitional exhaustion. Breath-by-breath measurements were taken throughout the protocol using a metabolic cart (Meta 3B, Leipzig, Germany) to establish VO_2_max, which was calculated as the highest average value over 30 s.

### 2.4. Measurements During Experiments

After randomization, subjects ingested BCAAs or the placebo for three consecutive days in the morning and evening. On the morning of the experimental trails, participants arrived at the laboratory after an overnight fast. First, fingertip blood and elbow vein blood were collected. Next, the participants ingested BCAAs or the placebo again and rested for 30 min. Elbow venous blood and fingertip blood were collected again 30 min after the treatment had been consumed. The participants then started to exercise. The subjects first conducted 1 h of CLE at 60% VO_2_max power on a power bike (Monark 839E, Vansbro, Sweden). During the CLE, participants were instructed to maintain a pedal speed between 60 and 70 RPM to minimize the impact of RPM variations on the experimental results. For 1 h CLE, RPE and VAS were recorded every 15 min, fingertip blood was collected to measure blood glucose and lactate every 15 min, HR was recorded every 10 min, and respiratory gas was collected to measure substrate metabolism every 10 min. The 80% VO_2_max power TTE was recorded immediately after CLE, encouraging subjects to reach truly subjective exhaustion. Gas was collected during all TTE periods and the exhaustion time was recorded after TTE. Elbow venous blood was collected immediately after exercise, and fingertip blood was collected to measure blood lactate post-exercise at 1, 5 and 9 min.

### 2.5. Diet Record and Supplementation Procedures

Throughout the experimental phase, subjects were required to keep dietary records. Subjects were asked to take photos of their daily diets to upload and record. The total energy intake and number of macronutrients consumed was analyzed using BoHe APP (Wuhan mint software V14.0.1, Wuhan, China) software. Subjects were asked to maintain the same diet as much as possible during both experiments (especially the day before both experiments) to avoid the influence of dietary factors on the results. An assessment of the subjects’ 3-day dietary records showed no significant differences between the two trail periods for energy and the analyzed nutrient intakes (Table 2).

Based on previous studies, we selected an appropriate average value for the supplementation dose, which was determined to be 0.2 g/kg body weight/each time for this study [24,26,30,32,33]. The BCAA supplementation ratio of leucine: isoleucine: valine was 2:1:1 (Beijing Competitor Sports Science &Tech. Co., Ltd., Beijing, China), and the placebo was isocaloric starch (Beijing Competitor Sports Science &Tech. Co., Ltd., Beijing, China). The supplement forms were both powders mixed with 400–450 mL of water, both of which have the same color, smell and taste.

### 2.6. Indirect Calorimetry Measurement

Respiratory exchanges (oxygen uptake [VO_2_], carbon dioxide production [VCO_2_]) were measured breath-by-breath through a face mask during the exercise tests using gas analyzers. During the cycling test on the CLE and TTE period, the VO_2_ and VCO_2_ values were collected to calculate substrate metabolism data. The CHO and fat oxidation rates were calculated according to the following equations [34]:Fat oxidation rate (g/min) = 1.695 × VO_2_ − 1.701 × VCO_2_CHO oxidation rate (g/min) = 4.21 × VCO_2_ − 2.962 × VO_2_

In the formula not corrected for protein oxidation, negative values for fat oxidation were corrected to 0. The area under the curve (AUC) was calculated by depicting the curve through the values at each time point.

The cycling efficiency was calculated using the average values recorded during TTE, using the following equations [35]:$$\mathrm{Energy}\;\mathrm{expenditure}\;(J\cdot s^{-1})=[(3.869\;\times \;{\mathrm{VO}}_{2})-(1.195\;\times \;{\mathrm{VCO}}_{2})]\;\times \;(\frac{4.186}{60})\;\times \;1000$$(1)$$\mathrm{Cycling}\;\mathrm{efficiency}\;(\%)=\frac{Power\;output(W)}{Energy\;expenditure\;(J\cdot s^{-1})}\;\times \;100$$

### 2.7. Perceptual Scales Assessment

The Borg 6–20 RPE scale was used to assess the participants’ perceived exertion, where a score of 6 indicates no exertion at all and 20 indicates maximal exertion. A visual analog scale was used to evaluate participants’ perceived level of muscle soreness. Soreness was assessed along a 10 cm scale (0 cm = no soreness, 10 cm = extreme soreness). Participants could drag the caliper to any point on a 0–10 cm scale based on their subjective muscle fatigue at the moment. The value was recorded to one decimal place.

### 2.8. Blood Samples Collection and Measurements

Electrochemical analyzers (EKF Biosen C-Line, Barleben, Germany) were used to analyze the subjects’ fingertip blood glucose and lactate concentrations. Venous blood was collected 5 mL at a time. Serum was collected using anticoagulation tubes containing sodium heparin. The aliquots were obtained by centrifugation at 3000× *g* at 4 °C for 10 min to separate serum and plasma and promptly stored in microtubes at −80 °C for laboratory measurements. Serum metabolites were analyzed for non-esterified fatty acids (NEFAs), insulin and blood ammonia using commercially available kits (Biosino Bio-Technology and Science Incorporation, Beijing, China) according to the manufacturer’s instructions.

### 2.9. Statistical Analysis

Sample size calculations were based on the primary outcome and fat oxidation, and calculated in G*power 3.1.9.7. Based on the effect size of fat oxidation, which was 0.4, an alpha of 5%, and a Power (1-β) of 0.8, the minimum number of participants required was 10. To account for dropouts, the number was increased to 12.

Data were analyzed using SPSS 26.0 (IBM^®^SPSS, Armonk, NY, USA) and statistical mapping was performed using GraphPad Prism 10.0 (GraphPad Software, San Diego, CA, USA). All data are expressed as mean ± standard deviations (SD). All outcomes were analyzed for normality (Shapiro–Wilk’s test). The subjects’ diet, TTE time, exercise efficiency and AUC area were analyzed using a paired *t*-test. The subjects’ HR, RPE, VAS, substrate oxidation rates, fingertip blood and venous blood during exercise were analyzed using two-way repeated measures ANOVA with time and treatment. Where a significant main effect was observed, pairwise comparisons were analyzed using Bonferroni post hoc tests in order to locate specific differences. Statistical significance was set at *p* < 0.05.

## 3. Results

### 3.1. BCAA Supplementation Promotes Fat Oxidation During Endurance Exercise

To investigate the effects of short-term BCAA supplementation on carbohydrate and lipid metabolism during exercise, respiratory gases were collected to calculate FATox and CHOox. The oxidation rates at different time points were used to construct curves, and the AUC was calculated. The fat oxidation rate showed a time effect (*p* < 0.001, η^2^ = 0.809) and interaction effect (*p* = 0.012, η^2^ = 0.126). The fat oxidation rate was significantly higher in the BCAA group than the placebo group at the 20th (*p* = 0.037) and 30th (*p* = 0.048) minutes of CLE (Figure 2a). The AUC of the fat oxidation rate was significantly higher for the BCAA group than the placebo group during CLE (29.91 ± 7.09 vs. 25.09 ± 8.85, *p* = 0.046, η^2^ = 0.688) (Figure 2b). However, there were no significant differences between the two groups in terms of the average fat oxidation rate (*p* = 0.394 and *p* = 0.95, respectively) and respiratory exchange ratio (RER) (*p* = 0.382 and *p* = 0.79, respectively) during CLE and TTE (Table 3). The data suggest that supplemental BCAA actually stimulates fat oxidation during endurance exercise, but the effect is shortly abolished by other metabolizing processes.

### 3.2. BCAA Supplementation Enhances CHO Oxidation During TTE

Next, we tested whether the carbohydrate oxidation increased and compensated for the fat oxidation after BCAA supplementation. We found that the CHO oxidation rate showed a time effect during CLE (*p* < 0.001, η^2^ = 0.87), but there was no main effect (*p* = 0.847, η^2^ = 0.002) and interaction effect (*p* = 0.906, η^2^ = 0.017) (Figure 2c). The AUC of the CHO oxidation rate was not significantly different between the two groups during CLE (72.53 ± 18.42 vs. 71.67 ± 15.00, *p* = 0.84, η^2^ = 0.062) (Figure 2d). However, we surprisingly observed that the average CHO at time to exhaustion (TTE) was remarkably increased (Table 3), indicating that a late response of CHO potentially compensates for the fat oxidation after BCAA digestion.

### 3.3. Blood Glucose and Lactate Are Not Affected by the BCAA Supplements

To further clarify which kind of carbohydrates are involved in the process, we first measured the blood glucose and lactate levels with or without BCAA supplements. The main effect of treatment (*p* = 0.24, η^2^ = 0.068) and interaction effect (*p* = 0.436, η^2^ = 0.047) on the blood glucose level were not significant, but blood glucose showed a time effect (*p* < 0.001, η^2^ = 0.193) (Figure 3a). The AUC of glucose was not significantly different between the two groups (*p* = 0.12). Meanwhile, the AUC of glucose at 0–15 min and 45–60 min did not show significant differences between the two groups (0–15 min: *p*= 0.13, 45–60 min: *p*= 0.08) (Figure 3b). The main effect of treatment (*p* = 0.848, η^2^ = 0.002) and interaction effect (*p* = 0.779, η^2^ = 0.021) on the lactate level were not significant, but lactate showed a time effect (*p* < 0.001, η^2^ = 0.549) (Figure 3c). The AUC of lactate was not significantly different between the two groups (*p* = 0.72) (Figure 3d).

### 3.4. NEFA Levels Were Not Affected by BCAA Supplements

To assess the effect of BCAA supplements on the free fatty acid metabolism of the blood, we collected venous blood samples and analyzed the non-esterified fatty acid (NEFA) levels (Figure 4). Due to special circumstances during the blood collection process and issues with sample contamination, only eight venous blood samples were available (NEFA, Insulin, Blood Ammonia) for the final analysis. Firstly, the exercise was found to alter blood NEFA levels significantly by showing a time-dependent effect (*p* < 0.001, η^2^ = 0.817). The pre-exercise NEFA levels were significantly lower than the fasting levels between the two groups (*p* < 0.01), and the post-exercise NEFA levels were significantly higher than the fasting (*p* < 0.001) and pre-exercise (*p* < 0.001) levels. However, BCAA supplements neither affected the levels of free fatty acids during endurance exercise (*p* = 0.252, η^2^ = 0.092) nor post-exercise (*p* = 0.977, η^2^ = 0.002). This means that the NEFA levels in the blood are not affected by BCAA supplementation (Figure 4).

### 3.5. Blood Insulin Was Elevated by BCAA Supplementation

Insulin is an important hormone that controls both fat and carbohydrate metabolism. We therefore tested whether the blood insulin levels were affected by the BCAA supplements during the endurance exercise. Interestingly, we found that the insulin levels showed a time effect (*p* < 0.001, η^2^ = 0.833) and interaction effect (*p* = 0.017, η^2^ = 0.253), and two significant alternations in the blood insulin levels were observed in the exercise experiments. First, the blood insulin level was elevated in post-exercise individuals regardless of BCAA supplementation. The post-exercise insulin level was significantly higher than pre-exercise between the two groups (*p* < 0.001). Second, the insulin level was significantly lower for the BCAA group than the placebo group post-exercise (*p* < 0.001) (Figure 5).

### 3.6. BCAAs Supplements Enhance Exercise Efficiency

To investigate the effect of short-term BCAA supplementation on exercise performance, respiratory gases during TTE were collected to calculate exercise efficiency, and the TTE duration was also recorded. We found that TTE showed no significant difference between the two groups (285.66 ± 77.27 s vs. 251.79 ± 97.89 s, *p* = 0.126, η^2^ = 0.504) (Figure 6a). Notably, cycling efficiency was significantly improved in the group with BCAA supplements (18.28 ± 1.77 vs. 17.45 ± 1.60%, *p* = 0.044, η^2^ = 0.695) (Figure 6b). These data exhibit the promising role of BCAAs in endurance exercise even though they were unable to extend the time to exhaustion.

### 3.7. BCAA Supplements Do Not Alter Physiological Responses During Endurance Exercise

Additionally, the subjects’ heart rate (HR) and rated perceived exertion (RPE) scale were measured to evaluate physiological function during the endurance exercise after short-term BCAA supplementation (Figure 7). BCAA supplements had no significant effect on HR (*p* = 0.882, η^2^ = 0.001) or an interaction effect (*p* = 0.888, η^2^ = 0.005), but HR showed a time effect (*p* < 0.001, η^2^ = 0.954) (Figure 7a). RPE also remained unchanged in addition to BCAA supplements during the time course (*p* = 0.381, η^2^ = 0.039) and no interaction effect was observed (*p* = 0.752, η^2^ = 0.023). However, RPE showed a time effect (*p* < 0.001, η^2^ = 0.917) that increased over time in the two groups (Figure 7b). The data suggest that the acute endurance exercise dictates the physiological responses rather than the BCAA supplementation.

### 3.8. BCAA Supplements Reduce Acute Exercise-Caused Fatigue

Finally, we evaluated the fatigue caused by exercise fatigue by calculating the visual analogue scale (VAS) and measuring the blood ammonia levels after BCAA supplementation (Figure 8). Remarkably, VAS was significantly higher in the group with BCAA supplementation than in the placebo group during the post-exercise period (5.77 ± 1.60 vs. 7.10 ± 1.28, *p* = 0.044) (Figure 8a). Meanwhile, blood ammonia also showed a time effect (*p* < 0.001, η^2^ = 0.866) and interaction effect (*p* < 0.001, η^2^ = 0.472). During the pre-exercise period, taking BCAA supplements increased blood ammonia levels significantly (*p* < 0.001). After exercise, the blood ammonia levels significantly increased in both groups compared to the pre-exercise levels (BCAA: *p* < 0.01; placebo: *p* < 0.001). Interestingly, the post- ammonia levels were significantly higher in the placebo group compared to the BCAA group (*p* < 0.05). (Figure 8b). Taken together, these data show that BCAA supplements can alleviate acute exercise-induced fatigue by increasing the ammonia in circulation.

## 4. Discussion

In this study, we have investigated the effects of short-term BCAA supplementation on substrate metabolism and exercise performance in active young males during an acute endurance exercise. We demonstrate that short-term BCAA supplementation enhances fat oxidation during moderate-intensity exercise and improves CHO oxidation during subsequent high-intensity exercise compared with a placebo. Although short-term BCAA supplementation does not increase the duration of TTE, it can enhance exercise efficiency during TTE. In addition, BCAA supplementation is associated with the alleviation of fatigue in the immediate post-exercise period. These findings show the importance of BCAA supplementation in exercise physiology in humans and provide evidence to support more diverse nutritional strategies during exercise.

We utilized the oxidation rate for the first time to explore the impact of BCAA intake on substrate metabolism during exercise. This approach provides a more intuitive visualization of data changes compared to the respiratory exchange ratio, RER. Consistent with our hypothesis, short-term BCAA supplementation enhances fat oxidation during endurance exercise. This is in line with previous findings of reduced RER during glycogen depletion exercises (80% of anaerobic threshold cycling to exhaustion) [30] and incremental loading exercises (starting at 50 W, the intensity was increased by 25 W every 2 min until exhaustion) [36] after the ingestion of BCAA. According to a previous review, two potential mechanisms have been proposed [12]: Firstly, the degradation of BCAA through the TCA cycle may simultaneously enhance both lipid synthesis and lipid oxidation, with the overall effect favoring increased energy expenditure; secondly, the oxidation of BCAA elevates acetyl-CoA levels and inhibits PDH activity, leading to a reduction in CHO oxidation and promoting fat oxidation. NEFA is a crucial biomarker of free fatty acid and influences exercise performance [37]. The pre-exercise NEFA levels in both groups were lower compared to fasting, and this aligns with previous studies [14]; this may result from the suppression of lipolysis by insulin secretion [38]. Interestingly, while differences in fat oxidation between the two groups were observed during endurance exercise based on gas exchange data, NEFA showed no significant difference post-exercise. This may be the result of both groups releasing large amounts of NEFA into the blood due to the TTE in the second phase [2].

This study found that the CHO oxidation rate was higher in the BCAA group compared to the placebo group during TTE. This is consistent with the results for fat oxidation. The intake of BCAA most likely enhances fat oxidation during CLE, preserves glycogen levels in the body, and thus increases CHO oxidation during subsequent TTE. This result agreed with an animal report stating that a four-week BCAA diet enables rats to maintain higher blood glucose and glycogen levels after exhaustive exercise compared to a placebo [11]. Additionally, the experiment found that rats consuming BCAA had reduced PDH activity, particularly in the liver. The inhibition of PDH activity can promote hepatic glycogen resynthesis and suppress CHO oxidation [39]. This is consistent with the previously mentioned potential mechanism by which BCAAs promote fat oxidation. The study found no difference in blood glucose between the two groups during CLE, which aligns with the lack of difference in CHO oxidation rates during this phase. Future studies are required to investigate the dynamic changes in blood glucose throughout the exercise period. Interestingly, lactate, which is not regulated by insulin, is also not affected by BCAA supplementation (Figure 4). Moreover, BCAA intake did not affect the pre-exercise insulin levels in the blood. For example, no change in insulin levels was observed in the BCAA group during pre-exercise, which contradicts the conclusion that leucine in BCAAs promotes insulin secretion [40]. Leucine’s effect on insulin may occur at a later point. A study on CHO combined with BCAAs and post-exercise muscle protein synthesis also observed similar results [31]. The insulin levels increased in both groups post-exercise compared to pre-exercise. This is due to a reduction in the muscle glycogen content post-exercise, which leads to increased glucose uptake and utilization, as well as enhanced insulin sensitivity [41]. In addition, insulin levels were higher in the placebo group than in the BCAA group post-exercise. The ability of exercise to reduce insulin secretion has been demonstrated in both animal [42] and human [43] experiments. Based on the results, it can be surmised that the placebo group consumed more glycogen during exercise, thus requiring more insulin secretion immediately post-exercise to enhance glucose uptake and muscle glycogen synthesis.

The fact that short-term BCAA supplementation significantly enhanced acute exercise performance in this study is notable. For instance, a significant increase in cycling efficiency was observed during the TTE stage (Figure 6). Previous studies involving BCAA combined with other nutritional supplements or supplementation throughout the entire exercise (load corresponding to 3 mmol of lactate) period have observed similar results [24]. Here, we show that the special function that BCAA supplementation offers may be a reason for the observed improvements. The improvement in exercise efficiency indicates lower energy expenditure during exercise and aligns with the finding that BCAA supplementation promotes CHO oxidation during TTE. However, although there was a numerical difference in TTE between the two groups, it did not reach statistical significance. However, the *p*-value shows a trend towards significance (*p* = 0.126). The potential reason for this could be the small sample size, and we hope to validate this result in future studies with a larger sample size (Figure 2).

Notably, BCAA supplementation is supposed to induce various physiological responses during exercise. These physiological responses are indicated mostly by HR and RPE. However, we find that BCAA supplementation does not alter both HR and RPE during acute endurance exercise (Figure 7). In fact, there are contradictory reports in the literature [24,27]. This may be because the impact of BCAAs on RPE depends on the duration of exercise.

One interesting observation is the reduction in fatigue caused by BCAA supplementation. This observation is consistently shown by VAS and the detection of blood ammonia (Figure 8). This experiment used VAS to assess the muscle fatigue levels during acute exercise and in the immediate post-exercise period. Compared to previous studies showing that BCAAs improve muscle soreness after eccentric exercise [44], a difference in VAS scores between the two groups was observed immediately after endurance exercise. An important factor contributing to post-exercise pain is the increased sensitivity of muscle receptors, which may be triggered by elevated levels of the inflammatory markers produced during exercise [45]. Glutamine, a catabolite of BCAA, has been shown to significantly reduce inflammatory responses in the body [46], which may explain the lower VAS scores observed in the BCAA group immediately post-exercise. In addition, the immediate fatigue experienced after endurance exercise may differ from the fatigue caused by eccentric exercise. This may be a result of neuromuscular fatigue induced by central fatigue due to prolonged exercise [47]. According to the central fatigue hypothesis [48] and the principle that BCAAs compete with 5-HT for entry into the brain [49], this could serve as another potential explanation for the immediate decrease in VAS post-exercise.

Besides VAS, blood ammonia elevation also explains the reason for BCAA-reduced fatigue. Elevated blood ammonia levels are believed to negatively affect central nervous system function, leading to phenomena such as impaired exercise control and drowsiness [50]. However, in this study, the increase in blood ammonia in the BCAA group did not negatively impact subsequent exercise (Figure 8). We suppose that a lack of BCAA supply during acute exercise leads to quick glycogen depletion and an increase in protein consumption for energy, eventually elevating blood ammonia. Moreover, as blood ammonia easily crosses the blood–brain barrier, this increase directly boosts the brain’s uptake of ammonia. It likely alters the brain’s energy metabolism and intraneuronal signaling pathways and causes central fatigue. Therefore, our findings, together with previous data, demonstrate that BCAAs play a role in alleviating central fatigue.

Despite the interesting results, we recognize that the lack of muscle biopsy is a major limitation of this study. This prevented us from observing changes in muscle glycogen and exploring the mechanism involved in the effect of BCAAs on substrate metabolism. Our study also focused on acute exercise with short-term supplementation, and we were unable to monitor long-term fatigue recovery post-exercise, nor assess the effects of prolonged BCAA supplementation on substrate metabolism and physiological responses during exercise. In addition, the small sample size may have prevented the observation of significant differences in some data, and the absence of female participants is also a limitation of this study. Future research could examine the gender differences in BCAA supplementation while increasing the sample size. Meanwhile, future studies that explore the effects of BCAAs combined with different nutritional supplements on exercise performance and metabolism are of interest. It is also worth investigating the beneficial physiological mechanisms of BCAA supplementation on exercise performance and metabolism through techniques such as biopsy or metabolomics.

## 5. Conclusions

This study highlights the function of short-term BCAA supplementation in acute endurance exercise in active young males (Figure 9). We find that BCAA supplementation enhances the fat oxidation rate during acute endurance exercise, also promoting the CHO oxidation rate and exercise efficiency during subsequent exhaustive exercise. Additionally, supplementation reduces post-exercise fatigue scores and increases blood ammonia levels, indicating its role in alleviating immediate post-exercise fatigue.

## Figures and Tables

**Figure 1 nutrients-17-01290-f001:**
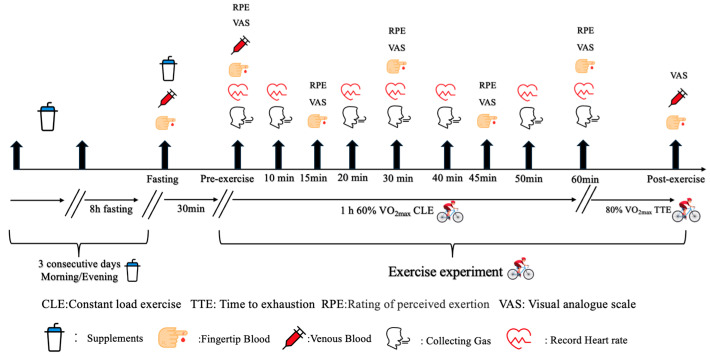
The figure depicts the framework of the study. The experiments started after three days of BCAAs supplemented twice per day. The endurance exercise was subsequently monitored by testing parameters such as blood, respiration and heart rates. After the exercise, the parameters were tested again to evaluate them post-exercise.

**Figure 2 nutrients-17-01290-f002:**
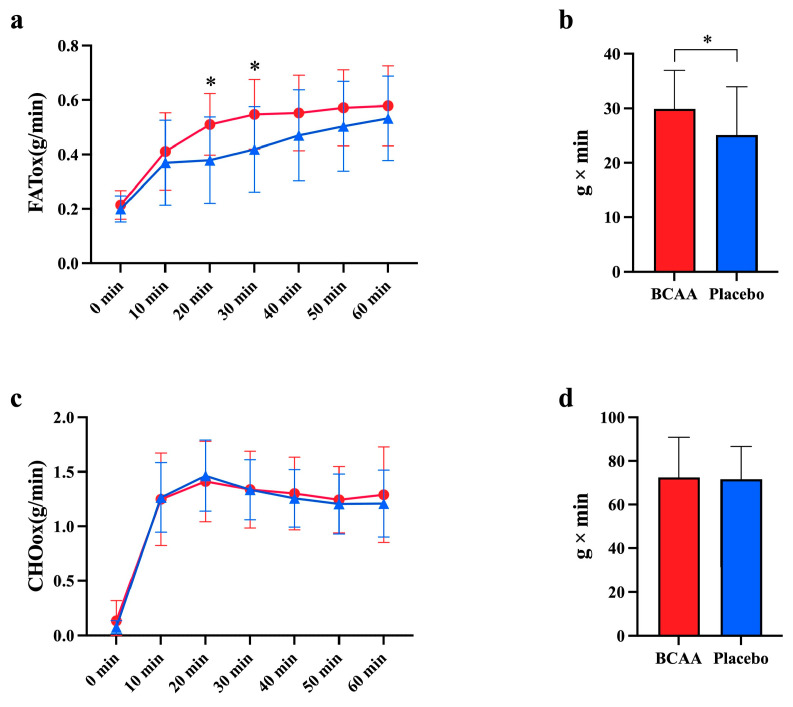
Fat and carbohydrate oxidation during exercise after BCAA supplementation. Respiratory gases were collected to calculate the oxidation rates marked FATox (**a**,**b**) and CHOox (**c**,**d**), respectively. * *p* < 0.05, *n* = 11. (**a**): Fat oxidation rate at different times during CLE, (**b**): The area under the curve (AUC) of fat oxidation during CLE, (**c**): CHO oxidation rate at different times during CLE, (**d**): The AUC of CHO oxidation during CLE. BCAA: Branched-chain amino acids.

**Figure 3 nutrients-17-01290-f003:**
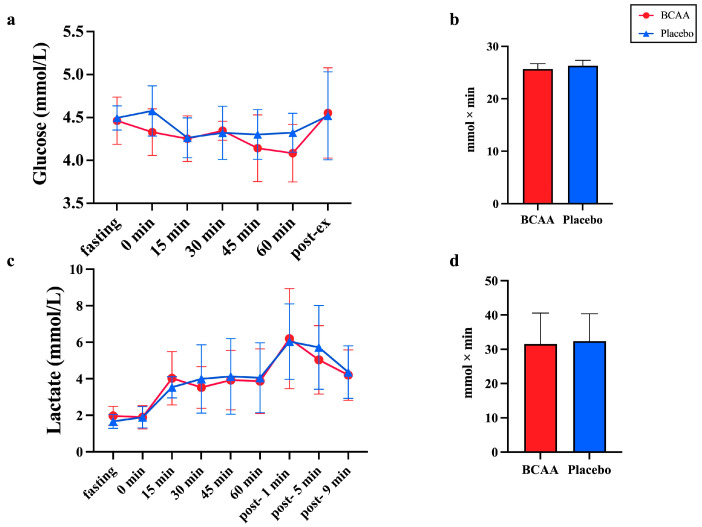
Blood glucose and lactate are not affected by BCAA supplements. The level of glucose (**a**,**b**) and lactate (**c**,**d**) during CLE and post-exercise. Values are means ± SD, *n* = 11. (**a**): Glucose level at different times, (**b**): Glucose AUC, (**c**): Lactate level at different times, (**d**): Lactate AUC. BCAA: Branched-chain amino acids.

**Figure 4 nutrients-17-01290-f004:**
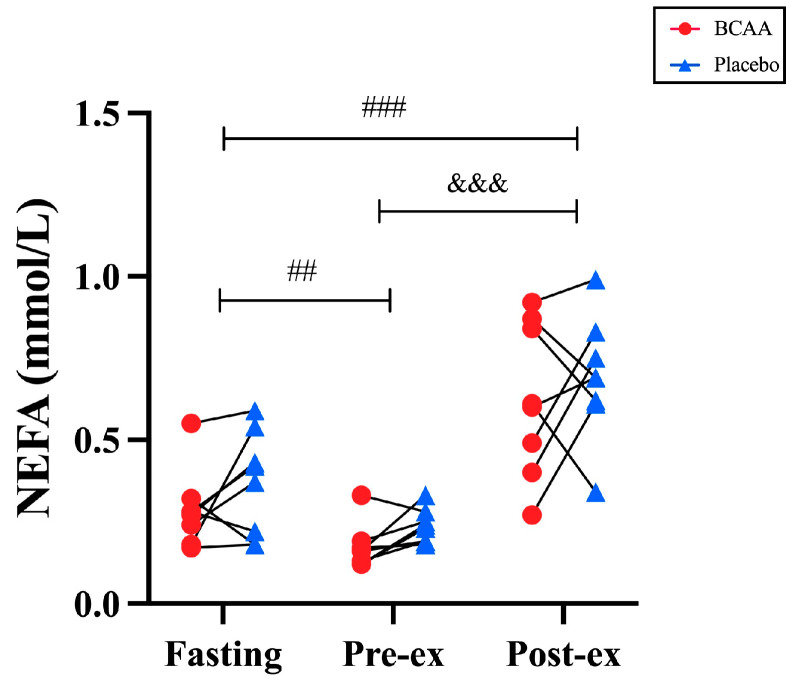
Non-esterified fatty acid (NEFA) levels in serum at fasting, pre-exercise and post-exercise. *n* = 8. ^&&&^ *p* < 0.001, vs. Pre-ex; ^##^ *p* < 0.01, ^###^ *p* < 0.001, vs. fasting. Pre-ex: pre-exercise, Post-ex: post-exercise.

**Figure 5 nutrients-17-01290-f005:**
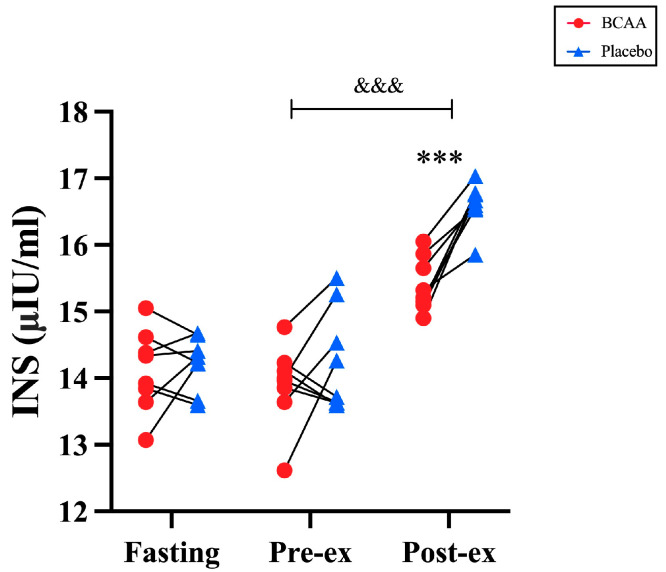
Blood insulin levels in serum at fasting, pre-exercise and post-exercise with BCAA supplements. *n* = 8. *** *p* < 0.001, ^&&&^ *p* < 0.001. Pre-ex: pre-exercise, Post-ex: post-exercise. INS: Insulin.

**Figure 6 nutrients-17-01290-f006:**
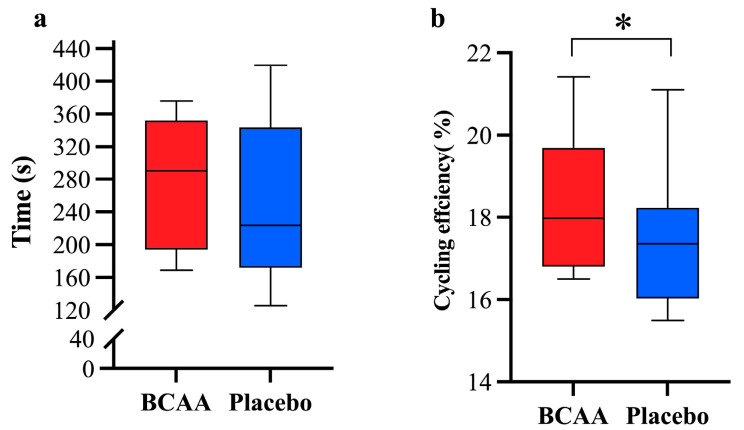
Time (**a**) and cycling efficiency (**b**) during TTE between two groups. *n* = 11. Values are means ± SD. * *p* < 0.05 vs. placebo group. TTE: time to exhaustion. BCAA: Branched-chain amino acids.

**Figure 7 nutrients-17-01290-f007:**
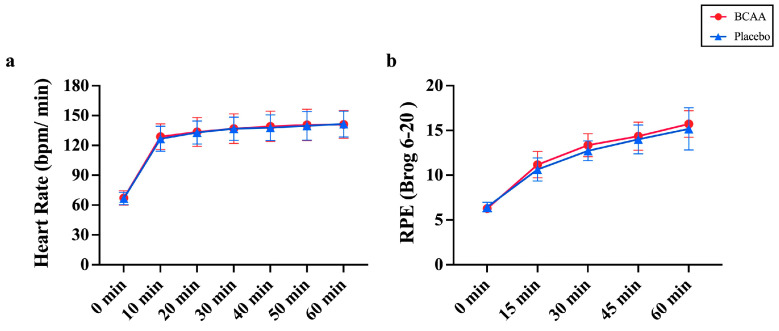
BCAA supplementation has no effect on heart activity during acute endurance exercise. (**a**) Heat rate (HR) was measured during 1 h of acute endurance exercise. (**b**) Rated perceived exertion (RPE) was detected at the same time. *n* = 11. Values are means ± SD.

**Figure 8 nutrients-17-01290-f008:**
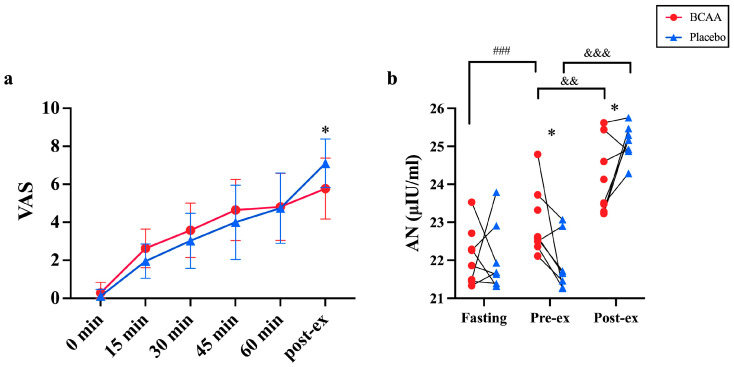
BCAA supplementation relieves acute exercise-induced fatigue. (**a**) VAS was measured during CLE. (**b**) Blood ammonia levels were detected in serum during fasting, pre-exercise and post-exercise. In (**a**), *n* = 11; in (**b**): *n* = 8. Values are means ± SD. * *p* < 0.05, ^&&^ *p* < 0.01, ^&&&^ *p* < 0.001, ^###^ *p* < 0.001.

**Figure 9 nutrients-17-01290-f009:**
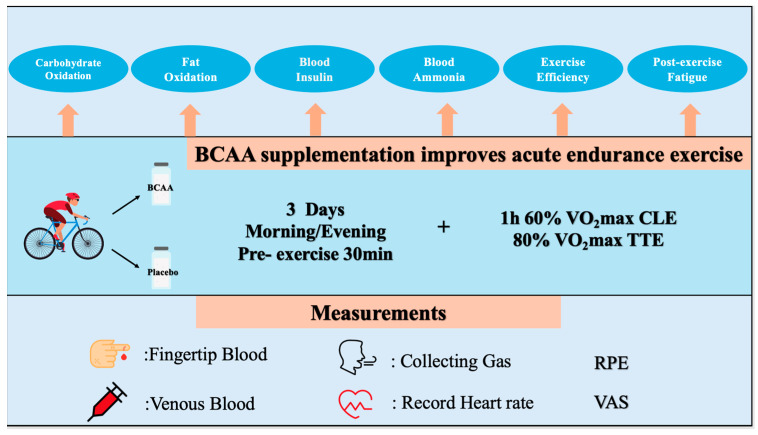
Conclusive figure. The figure shows an overview of the experiments.

**Table 1 nutrients-17-01290-t001:** Participant characteristics.

Characteristics	Mean ± SD
Age (year)	21 ± 1
Height (m)	1.79 ± 0.08
Body mass (kg)	74.8 ± 10.3
Body fat (%)	16.7 ± 5.4
BMI (kg/m^2^)	23.5 ± 1.9
VO_2_max (mL/(kg·min))	49 ± 9
power VO_2_max (w)	228 ± 28
power at 60% VO_2_max (w)	117 ± 21
power at 80% VO_2_max (w)	170 ± 25

**Table 2 nutrients-17-01290-t002:** Dietary intake from 3-day records at each precondition period.

	BCAA	Placebo
Energy (kcal)	2062 ± 338	2076 ± 348
Carbohydrate (g)	246 ± 50	246 ± 47
Fat (g)	64 ± 18	67 ± 17
Protein (g)	115 ± 11	116 ± 15

BCAA: Branched-chain amino acids.

**Table 3 nutrients-17-01290-t003:** Average of metabolic variables during CLE and TTE in the BCAA and placebo trails.

	CLE	TTE
**FATox (g/min)**		
BCAA	0.51 ± 0.14	0.18 ± 0.18
Placebo	0.48 ± 0.16	0.19 ± 0.23
**CHOox (g/min)**		
BCAA	1.33 ± 0.30	3.20 ± 0.87 *
Placebo	1.35 ± 0.22	2.79 ± 1.04
**RER**		
BCAA	0.85 ± 0.03	0.98 ± 0.05
Placebo	0.86 ± 0.03	0.98 ± 0.07

Note: *n* = 11. Values are means ± SD. * *p* < 0.05, vs. placebo group. CLE: constant load exercise; TTE: time to exhaustion; FATox: fat oxidation rate; CHOox: carbohydrate oxidation rate; BCAA: Branched-chain amino acids; RER: respiratory exchange ratio.

## Data Availability

All data and materials of this study are available upon request due to ethical reasons.

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
