# Peer review of "Branched-Chain Amino Acid Supplementation Enhances Substrate Metabolism, Exercise Efficiency and Reduces Post-Exercise Fatigue in Active Young Males"

_nutrients, 2025, doi:10.3390/nu17071290_

Round 1
Reviewer 1 Report
Comments and Suggestions for Authors
Title. I suggest to change “Branched Chain” to “Branched-Chain”.
Title. I suggest to change “Exercise Efficiency during Exercise” to “and Exercise Efficiency”.
It is problematic that the rpm was not controlled during the CLE and TTE. An inconstant rpm during TTE may affect cycling efficiency and substrate oxidation. The authors need to provide the rpm data during the CLE and TTE.
In the abstract, the conclusion needs to be revised and more specific. It is stated too general.
Please justify why the study was not done in cyclists.
L29. “CHO oxidation rates also showed a significant increase in the BCAA group(p<0.05)”. This did not happen during the CLE. Please revise.
L32. “Additionally, the levels of blood insulin were significantly higher during both fasting and pre-exercise periods (p<0.001)”. That is in contract with the data presented in the results section. Please clarify.
L47. Ref 1 is a review article on primarily glycogen and carbohydrate not on substrate metabolism. Please replace ref 1 and provide primary sources that justify “subtle changes” that enhance exercise performance. In addition, subtle is vague so I suggest to be more specific.
L56. Please delete “external”.
L62. It is stated “produce large amounts of adenosine triphosphate”. Is that really the case? Please reconsider the statement.
L79. Ref 17 was in liver cirrhosis patients. Could BCAA enhance muscle strength in non-clinical cohorts?
L100. “improves CHO oxidation”. That is not presented in the data. Please revise.
Ls 104-105. Please reconsider “Therefore, this study provides potential strategies for developing BCAA-based therapeutics for patients with impaired exercise capacity and substrate metabolism.”. the study has healthy participants.
In the manuscript I suggest to clarify when you provide a reference of a review.
L110. How many participants were recruited who did not meet the guidelines for physical activity? Please clarify.
Table 1. Please express mean and SD values of age, body weight, VO2max and power values without decimal places. Body fat and BMI can be expressed with one decimal place.
L137. It seems testing was in the morning after an 8 hr fast. I suggest to clarify that here.
L139. Please replace “supplementals” with “supplementation”.
L144. What was measured with the VAS. Please clarify.
In the methods section, please provide a rationale for the dosing strategy. i.e. dose, intake duration and timing of intake.
Table 2. Please express mean and SD values in Table 2 without decimal places.
L171. For the 60% and 80%VO2max testing, why was the rpm not controlled. Please provide the rpm values as different values may result in different motor-unit recruitment and affecting substrate metabolism.
L192. “Synthesizing previous studies”, but only one reference is provided. Please revise. In addition, in ref 26 the participants consume three times per day the supplement. Please provide justification for the dosing strategy as already mentioned above.
L242. P=0.12 so the interaction was not significant. Please revise/clarify.
Figures 2 and 3. Area has % units. Please clarify.
Figures 3a and c. I suggest to have provide also glucose values at identical recovery time points as lactate.
L294. “This means the NEFA are excluded in the increased respiratory fat oxidation by BCAA supplements”. How is this physiologically possible. What is the mechanism? This statement needs to be explained.
Figures 4 and 5. There seems to be only data of 8 participants? Please clarify.
Ls 307. The following statement needs revision “Second, BCAA supplements enhanced the insulin levels in the individuals at the post-exercise stage and the insulin level was significantly lower for BCAA than placebo in the post-exercise (p<0.001) (>”. as enhanced and lower is in contrast. In addition, L309 needs revision as well as BCAA lowered insulin.
L319. Was the study underpowered as BCAA TTE seems to be ~13% higher. A 13% change in performance time would be meaningful and sport nutrition studies that cannot show a 13% change as significant are problematic. This needs to be discussed.
L360. What is an “acute endurance exercise”. Is it not just “endurance exercise”?
L363. There was no improvement in TTE. Please revise.
L368. What would you need to develop a drug when you suggest the beneficial effects can be due to BCAA intake? Please revise.
L375. Throughout the manuscript and when referring to other studies with BCAA, please provide for those studies the dosing strategy, exercise modality and participant info. For example, for ref 34 it just mentions “incremental loading exercise”. That is not very informative.
L378. “enter the TCA cycle, potentially enhancing lipid synthesis and subsequently increasing energy expenditure and fat oxidation;” This is akward as the TCA cycle is not beta-oxidation. In addition, lipid synthesis would not really happen during exercise. Please revise.
L382. Please clarify what time point you are referring to regarding NEFA levels?
L382. “The change in NEFA levels is observed in both groups after BCAA or placebo intake”. There was no change between conditions at identical time points. Please revise.
L388. NEFA use by the respiratory system is problematic. Please revise.
L391. Insulin controlling the endocrine system is problematic. Please revise.
L393. Increased hepatic glucose oxidation would lower glucose levels. This is problematic.
L399. BCAA cannot regulate insulin levels. This is problematic. Please revise.
L432. There is mention here of muscle soreness which was likely delayed onset muscle soreness which is the result of very different mechanisms than fatigue reported in the present study.
Ls 450. Why is the participant description linked with the inability to get biopsies. Please revise.
Comments on the Quality of English LanguageSometimes, the grammar is problematic and may have affected contrasting statements that are present in the manuscript.
Author Response
Dear Reviewer,
We sincerely appreciate your time and effort in reviewing our manuscript. Your insightful comments and constructive suggestions have been extremely valuable in improving the quality of our work. We have carefully considered each point and made the necessary revisions accordingly.The specific response will be sent to you as a Word document attachment.
Thank you again for the time and effort you have spent reviewing this manuscript and for your valuable suggestions. They have undoubtedly enhanced the quality of our article. We look forward to your response.
Your sincerely,
Chenglin Luan
March 23, 2025

Reviewer 2 Report
Comments and Suggestions for Authors
In this study, authors reported that BCAA enhances substrate metabolism during exercise in active males and improves exercise efficiency and alleviates exercise-induced fatigue. Nutritional support during exercise must be important for safe and effective exercise. Although there are several limitations in this study, their findings could give us new insight into the relationship between exercise ad nutrition.
They should include some discussion or limitations about the points below:
1. Although they conducted cross-over trial, Limited sample size (n = 11) may reduce the generalizability of the findings.
2. Short-term supplementation (only three days) does not allow for long-term conclusions about BCAA effects.
3. They include exclusively male participants—findings may not be applicable to female population.
4. Lack of information about synergistic effects of BCAA with other nutrients (e.g., carbohydrates, lipid etc.) on endurance performance and fatigue reduction.
Author Response
Dear Reviewer,
We sincerely appreciate your time and effort in reviewing our manuscript. Your insightful comments and constructive suggestions have been extremely valuable in improving the quality of our work. We have carefully considered each point and made the necessary revisions accordingly.
Below, we provide detailed responses to your comments, highlighting the modifications made in the manuscript.
Reviewer’s comments:
In this study, authors reported that BCAA enhances substrate metabolism during exercise in active males and improves exercise efficiency and alleviates exercise-induced fatigue. Nutritional support during exercise must be important for safe and effective exercise. Although there are several limitations in this study, their findings could give us new insight into the relationship between exercise ad nutrition.
They should include some discussion or limitations about the points below:
- Although they conducted cross-over trial, Limited sample size (n = 11) may reduce the generalizability of the findings.
2. Short-term supplementation (only three days) does not allow for long-term conclusions about BCAA effects.
3. They include exclusively male participants—findings may not be applicable to female population.
4. Lack of information about synergistic effects of BCAA with other nutrients (e.g., carbohydrates, lipid etc.) on endurance performance and fatigue reduction.
Responds to common:
Thank you for your valuable feedback regarding the limitations of our study. Following your suggestion, we have added a discussion on the limitations of this study in the discussion section.
Thank you again for the time and effort you have spent reviewing this manuscript and for your valuable suggestions. They have undoubtedly enhanced the quality of our article. We look forward to your response.
Your sincerely,
Chenglin Luan
March 23, 2025
Reviewer 3 Report
Comments and Suggestions for Authors
This study is interesting in the context of fatigue, which is currently increasing.
Introduction
There is a gap between the two issues, which I believe should be further discussed. The first ventilatory threshold was 70% VOmax, and the intensity of effort was not excessive, unless the subject had a low level of training. It is necessary to observe the reaction at different intensity thresholds to obtain clear references.
The central level fatigue appears mainly at high intensities, being peripheral in lower work, so the response is different, please clarify this detail.
Methods
A small sample size should be considered in this study.
This fact suggests that the authors changed the title of the study by adding that it is a pilot study, as well as the fact that it does not include a female sample.
It should be indicated whether the training habits (not the study procedure) are similar or not, and the rate of recovery will vary.
Discussion
Please tone down the achievements due to the small sample size, which does not allow for the generalization of the data.
There is a lack of emphasis on the practical applications of the exercise and perhaps naming it according to the VOmax requirement.
Please add a separate section regarding these limitations.
Author Response
Dear Reviewer,
We sincerely appreciate your time and effort in reviewing our manuscript. Your insightful comments and constructive suggestions have been extremely valuable in improving the quality of our work. We have carefully considered each point and made the necessary revisions accordingly. The specific response will be sent to you as a Word document attachment.
Thank you again for the time and effort you have spent reviewing this manuscript and for your valuable suggestions. They have undoubtedly enhanced the quality of our article. We look forward to your response.
Your sincerely,
Chenglin Luan
March 23, 2025

Round 2
Reviewer 1 Report
Comments and Suggestions for Authors
Thanks for your replies to my comments. I still have a few.
Please clarify throughout the manuscript when data is presented from 8 participants, i.e. in the abstract, figure legends etc.
Ls 29-30. According to information in the results section (L279), there was no change in carbohydrate oxidation. Please revise. See also L105 “improves CHO oxidation” seems to be incorrect.
L124. “The final sample comprised thirteen young males (Table 1)”. This is incorrect. Please revise.
Table 1. Please change “VO2max watt (w)” to “Power VO2max (W). Make those changes as well for 60% and 80%. In addition, note that Watt is the unit and power is the parameter, see e.g. L133, change “60% and 80% of VO2max watt” to “the power at 60% and 80% of VO2max”. Make changes throughout the manuscript as required.
L155. Change “carton” to “figure”.
L209. Please revise/clarify “During the cycling test on CLE and TTE period, the VO2 and VCO2 values were analyzed to obtain values”.
L254. Note that carbohydrate oxidation is not only glucose but also glycogen. Please revise.
Figures 2b and 2d. Please clarify what is plotted here. Is this the total amount of fat oxidized during the hour?, or is it AUC.
You had venous blood collection but in your letter you mention “Some participants were unable to tolerate arterial blood collection”. What was done? Please clarify.
L335. The SDs here in the text for cycling efficiency do not seem to match with those in the figure. In the placebo condition, the SD is smaller compared to BCAA but larger in the Figure. Please clarify.
L364. Are you sure that “Similarly, at the post-exercise stage, blood ammonia was also raised after taking the BCAA supplements (p<0.001) (Figure 8b)><0.001) (Figure 8b).” It look like the BCAA average ammonia was lower than placebo. Please check.
L408. “The high blood glucose observed in the placebo group during this period may result from muscle glycogen breakdown contributing to blood glucose”. This needs revision as breakdown of muscle glycogen does not contribute to changes in blood glucose.
Comments on the Quality of English LanguageSometimes, the grammar is problematic.
Author Response
Dear Reviewer,
Thank you very much for taking the time to review our revised manuscript and providing us with valuable feedback once again. We sincerely appreciate your thoughtful comments, which have helped us improve the quality of the manuscript. We have carefully addressed the points you raised and have made further revisions accordingly. The detailed responses and revisions have been provided in the attached Word document. Please find the attachment for further details.
Thank you again for your constructive comments and support.We believe the revisions have strengthened the manuscript, and we are eager to hear your thoughts on the updated version. We look forward to your response
Your sincerely,
Chenglin Luan
March 28, 2025

Round 3
Reviewer 1 Report
Comments and Suggestions for Authors
Thanks for addressing all my comments.